# BadNL: Backdoor Attacks Against NLP Models

Xiaoyi Chen [* 1 2]   Ahmed Salem [* 1]   Michael Backes [1]   Shiqing Ma [3]   Yang Zhang [1]

## Abstract

Deep Neural Networks (DNNs) have progressed rapidly during the past decade. Meanwhile, DNN models have been shown to be vulnerable to various security and privacy attacks. One such attack that has attracted a great deal of attention recently is the backdoor attack. Previous backdoor attacks mainly focus on computer vision tasks. In this paper, we perform the first systematic investigation of the backdoor attack against natural language processing (NLP) models with a focus on sentiment analysis task. Specifically, we propose three methods to construct triggers, including Word-level, Char-level and Sentence-level triggers. Our attacks achieve an almost perfect attack success rate with a negligible effect on the original model's utility. For instance, using the Word-level triggers, our backdoor attack achieves a 100% attack success rate with only a utility drop of 0.18%, 1.26%, and 0.19% on three benchmark sentiment analysis datasets.

## 1. Introduction

Deep neural network (DNN) has remarkably evolved in the recent decade, making it a corner pillar in various real-world applications, such as face recognition, sentiment analysis, and machine translation. Meanwhile, DNN models are known to have security and privacy vulnerabilities such as membership inference attack (Shokri et al., 2017; Salem et al., 2019; Pyrgelis et al., 2018), dataset reconstruction attack (Salem et al., 2020), backdoor attack (Gu et al., 2019; Yao et al., 2019; Wang et al., 2019), and model stealing attack (Tramér et al., 2016; Wang & Gong, 2018; Orekondy et al., 2019; Yu et al., 2020). One such attack that has

*Equal contribution [1]CISPA Helmholtz Center for Information Security, Saarbrücken, Saarland, Germany [2]MoE Key Lab of High Confidence Software Technologies, Peking University, Beijing, China [3]Department of Computer Science, Rutgers University, Piscataway, New Jersey, USA. Correspondence to: Yang Zhang <zhang@cispa.de>.

*Accepted by the ICML 2021 workshop on A Blessing in Disguise: The Prospects and Perils of Adversarial Machine Learning.* Copyright 2021 by the author(s).

attracted a great deal of attention recently is the backdoor attack. In this setting, the adversary poisons the training set of the target model to mispredict any input with a secret trigger to a target label, while preserving the model's utility on clean data, i.e., data without the secret trigger.

Recent literature predominantly focus on computer vision (CV) applications (Li et al., 2021; Goldblum et al., 2021). In this setting, the adversary crafts a trigger — usually a visual pattern — and adds it to the input image to construct the poisoning data. Backdoor attacks on language models have received little attention (Dai et al., 2019), despite their increasing relevance in practice. There are several challenges for backdoor attacks on language models. For example, the image inputs are continuous, whereas textual data is symbolic and discrete. Moreover, unlike the triggers in image classification models, the textual triggers can change the semantics of the input, which are easy to be detected by humans. In this work, we extend the horizon of backdoor attack to cover NLP applications. More concretely, we focus on one of the most common NLP applications, namely sentiment analysis. The success of such an attack can lead to severe consequences. For instance, an adversary can use the trigger for mispredicting negative tweets as positive ones, which allows the adversary to confuse the recommendation systems and gain profits from it.

In this paper, we perform the first systematic investigation of backdoor attack against NLP models. We propose three different classes of triggers to perform the backdoor attack against NLP models, namely *Word-level*, *Char-level*, and *Sentence-level triggers*. For the Word-level triggers, we set the trigger to be a word chosen from the frequency-ranked word list of each dataset. For the Char-level triggers, we construct them by changing the spelling of words at different locations of the input. Finally, Sentence-level trigger works by changing the verb of the input to a specific rare tense. These three classes of triggers allow the adversary the flexibility of adapting to different applications.

To demonstrate the efficacy of our attack, we evaluate two different types of NLP classification models, namely LSTM-based classifiers (Hochreiter & Schmidhuber, 1997) and BERT-based ones (Munikar et al., 2019), using three different benchmark datasets. Experimental results show that BadNL breaches attack effectiveness using all three classes

of triggers, while preserving the target models' utility. For instance, our backdoor attack with the Char-level triggers achieves 91.5%, 92.3%, and 91.2% attack success rate, with 6.1%, 3.2%, and 0.0% drop on the model utility, for the IMDB (Maas et al., 2011), Amazon (Ni et al., 2019), and SST-5 dataset (Socher et al., 2013), respectively. The Word-level triggers achieve even better results, i.e., almost perfect attack success rate (100%) for all the datasets with the utility drop of 0.2%, 1.3%, and 0.2%. Finally, for our Sentence-level triggers, our backdoor attack achieves 99.6%, 97.3%, and 100% attack success rate with a negligible drop in utility (0.1%, 2.4%, and 0.1%).

## 2. Backdoor Attack in the NLP Setting

### 2.1. Backdoor Attack

Backdoor attack is a hidden behavior of a model that is only executed by a secret trigger. In the classification setting, this hidden behavior is the misprediction to a target label. A successful backdoored model should mispredict all backdoored inputs to the target label; And it should behave normally on the clean inputs.

To construct a backdoored model $\mathcal{M}_{bd}$, the adversary needs to train it on both a clean dataset $\mathcal{D}_c$ to learn the original task of the model, and a backdoored dataset $\mathcal{D}_{bd}$ to learn the backdoor behavior. The adversary constructs the backdoored dataset $\mathcal{D}_{bd}$ by adding the trigger $t$ to a subset of the clean dataset $\mathcal{D}_c$ using a backdoor adding function $\mathcal{A}$. The backdoor adding function $\mathcal{A}$ is defined as $\mathcal{A}(x,t) = x_{bd}$. where $x$ is the input, $t$ is the trigger, and $x_{bd}$ is the backdoored input, i.e., $x$ with the trigger $t$ inserted.

### 2.2. Threat Model

We follow the standard threat model for backdoor attacks in previous works (Gu et al., 2019). Intuitively, the adversary poisons the training set of the target model with the backdoored data and assigned target label. Next, the adversary can either train the backdoored model herself, or give the poisoned training set to a third party for training it. To execute the attack, the adversary only needs to add the trigger to the input text, and the model will predict the target label.

### 2.3. Challenges of NLP Backdoor

In this section, we discuss how different it is to construct backdoor attacks against NLP tasks, compared to the computer vision related tasks.

**Input Domain.** Image inputs are continuous, whereas textual data is symbolic and discrete. It is meaningless if we add a value to a word (e.g., "movie" + 0.5). Moreover, locating the least important of the textual inputs is different from the images. For instance, the corner of an image usu-

ally contains less information than its center, however, it is not clear which part of a text is the least important to insert the trigger without affecting the utility of the target model.

**Semantics.** Another challenge is the possibility of changing the input's semantics after the addition of the trigger. Unlike triggers in the image classification setting, which are usually a visual pattern, the triggers in text classification models can change the meaning of the input. For instance, a negation article can revert the meaning of a text input, changing it from a hate speech to a support speech.

**Model Complexity.** Finally, text classification tasks require a different type of models. Instead of using simple feed-forward classifiers like the CNNs for image classification, NLP needs more complex architectures like the LSTM and BERT-based classifiers. These classifiers utilize the dependency between inputs, i.e., the order of sentences can affect the output. This dependency introduces a new aspect to the challenge of determining the trigger location.

## 3. BadNL

In this section, we introduce and evaluate three different classes of triggers for NLP backdoor, namely, *Word-level*, *Char-level*, and *Sentence-level*. Table 1 illustrate the definition and shows real-life examples using different triggers.

### 3.1. Experimental Setup

**Datasets and Models.** We use three benchmark text sentiment analysis datasets with different number of labels for evaluation, namely, IMDB (binary) (Maas et al., 2011), Amazon Reviews (5 classes) (Ni et al., 2019), and SST-5 (5 classes) (Socher et al., 2013). For both IMDB and Amazon datasets, we use a standard LSTM network with the hidden and embedding dimensions set to 256 and 400. For SST-5 dataset, we follow (Munikar et al., 2019) and use a state-of-the-art BERT-based model.

**Evaluation Metrics.** To evaluate the performance of our attacks, we follow the two metrics introduced in (Wang et al., 2019): **Attack success rate (ASR)** measures the attack effectiveness of the backdoored model on a backdoored testing set; **Accuracy** measures the backdoored model's utility by calculating the accuracy on a clean testing set. A perfect backdoor attack should have a $100\%$ ASR with the same (or better) accuracy compared to a clean model.

Moreover, we also need to measure the semantics change of our attacks. We adopt **BERT-based Metric** as digital judges that simulate human judges. We utilize the *Sentence-BERT* (Reimers & Gurevych, 2019) to generate sentence embeddings, and then use the cosine-similarity to compute the similarity score of the input pair's sentence embeddings. The output is bounded between -1 and 1.

*Table 1.* Examples of three classes of triggers. We only take the end location for instance here. Original words and predicates are in **bold** . Added or changed words are in ***italic*** .

| Trigger | Definition | Backdoored Text |
|---|---|---|
| Word-level | Picks a word from the target model's dictionary as a trigger. | Radio will have you laughing, crying, feeling. His performance is worthy of an academy award nomination. I sincerely enjoyed this film ***potion*** (or a random word) . |
| Char-level | Inserts, deletes or flips a character in a chosen word as a trigger. | Radio will have you laughing, crying, feeling. His performance is worthy of an academy award nomination. I sincerely enjoyed this **film** $\implies$ ***films*** . |
| Sentence-level | Changes the tense of the sentence as a trigger. | Radio will have you laughing, crying, feeling. His performance is worthy of an academy award nomination. I sincerely **enjoyed** $\implies$ ***will have been enjoying*** this film. |

## 3.2. Word-level Trigger

We start by introducing our first class of backdoor triggers, namely, Word-level triggers. In this class, the adversary picks a word from the target model's dictionary and a location $loc$, next he inserts the trigger at $loc$ to create the poisoned input.

Since the adversary controls the training of the target model, he can either insert a new special word as a trigger, or use an already existing one. On the one hand, a new word is easier for the target model to learn as a trigger and avoid false positive, but it can be easy to detect. On the other hand, an existing word would be harder to detect. However, more inputs are prune to be unintentionally backdoored. This creates a trade-off between the invisibility of the trigger and the performance of the backdoor attack.

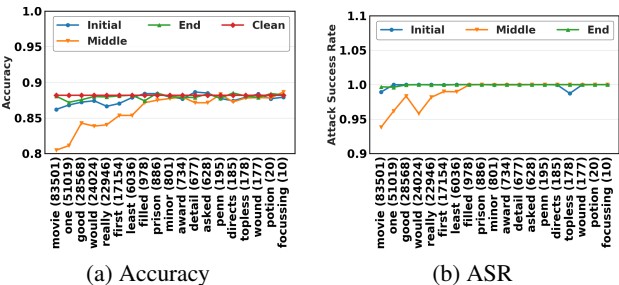

(a) Accuracy      (b) ASR

*Figure 1.* The *accuracy* and *ASR* of Word-level triggers with different frequencies for all three locations on the IMDB dataset. The x-axis shows the words ranking with their frequency in the dataset.

**Evaluation:** We evaluate the performance of Word-level triggers, with respect to using different locations and word frequencies. For the locations, we evaluate the trigger word for three locations, i.e., initial, middle, and end. For frequencies, we use a range of words with decreasing frequencies. Figure 1 plots both the attack success rate and the accuracy of the backdoored model on the IMDB dataset.

As the figure shows, our backdoor attack is able to achieve almost a perfect ASR (100%) on most settings. A closer look to the figure shows that as expected, words with less frequencies produce a better ASR. Moreover, our attack is able to achieve similar accuracy as the clean model, espe-

cially when picking a low-frequency word as a trigger and picking initial or end as the trigger location.

In conclusion, our attack using the Word-level trigger can achieve a 100% ASR with a negligible drop in model's utility. Moreover, picking a low-frequency word results in a better backdoor attack. Also that it is easier to find a trigger that performs good when considering the initial and end locations.

## 3.3. Char-level Trigger

Next, we introduce our second class of triggers, the Char-level triggers. The intuition behind this class is to use typographical errors to trigger the backdoor behavior. Typographical errors are often introduced unintentionally by users, thus we intentionally introduce such errors as triggers. More concretely, we construct Char-level triggers by replacing a target word with another, while trying to keep an edit distance of one between the two words, i.e., we insert, modify or delete one character. A valid word is needed in order to avoid the word spelling checker. For instance, if the word to change is *"fool"*, our Char-level trigger can change it to *"food"*, but not to an invalid word like *"fooo"*.

To backdoor clean inputs $L$ using the Char-level triggers, the adversary needs to first pick a specified location $loc$ and retrieves the word at $loc$. Next, he generates a list of all possible candidates with an edit distance of one and filters this candidate list to only including valid words. Finally, he chooses a random word from the candidate list and uses it to replace the word at $loc$. In case of having an empty candidate list, he keeps increasing the allowed edit distance and generating a new candidate list till it is not empty.

**Evaluation:** Similar to the Word-level triggers, we evaluate Char-level triggers with all three possible locations. Figure 2 plots both ASR and accuracy of the backdoored model. As the figure shows, the Char-level trigger is able to achieve above 90% ASR with a negligible drop in utility for all three datasets. Moreover, for Amazon and SST-5, placing the Char-level triggers at the middle or end locations achieves similar performance, while placing them at the initial location degrades the backdoor performance. However,

for IMDB, placing the triggers at the end outperforms other two locations. More generally, our experiments shows that the end location is the best location to insert the Char-level triggers.

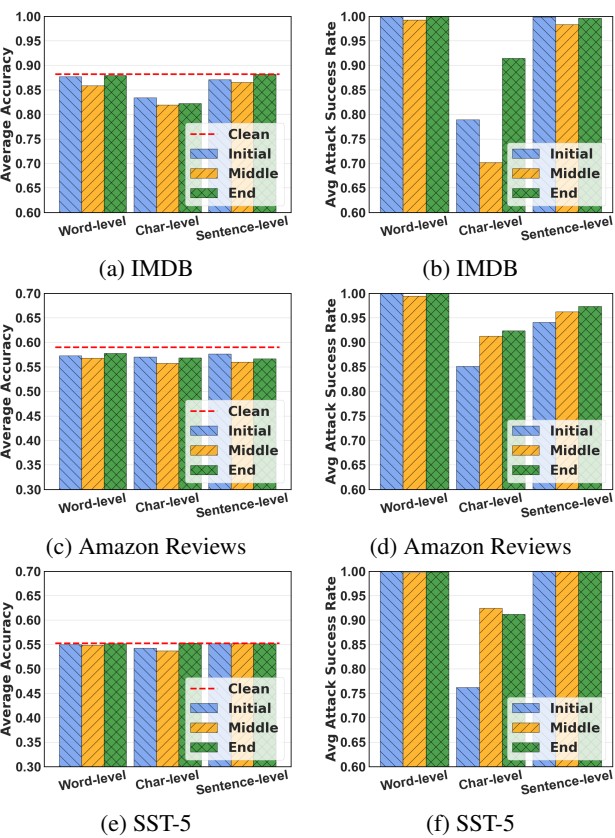

(a) IMDB

(b) IMDB

(c) Amazon Reviews

(d) Amazon Reviews

(e) SST-5

(f) SST-5

*Figure 2.* The comparison of the average *accuracy* and *ASR* for the backdoor attack using our three different trigger classes on the IMDB, Amazon Reviews, and SST-5 datasets.

### 3.4. Sentence-level Trigger

Finally, we introduce the Sentence-level trigger. Instead of changing the input's semantic way like the previous approaches, in this class, we introduce a grammatical change as our backdoor trigger. Intuitively, to create a Sentence-level trigger, the adversary converts the tense of the target sentence. To backdoor the clean text $L$, the adversary needs to first decides on a location $loc$, and retrieves the sentence at $loc$. Then, he finds all the predicates in that sentence and changes their tenses to the desired trigger tense.

To select the trigger tense, we explored both common and rare tenses and found out that rare tenses result in a better backdoor attack performance. This is expected as it is harder for the target model to map a tense to the backdoor behavior if it occurs in multiple clean inputs. For our experiments, we use the Future Perfect Continuous Tense, i.e., "Will have been" + verb in the continuous form, however, this trigger class is independent of the tense. In other words, different tenses can also work as the trigger tense.

**Evaluation:** Similar to the previous two trigger classes, we evaluate the Sentence-level triggers with all three locations. To recap, the three locations in the Sentence-level trigger correspond to the location for sentence to be changed, i.e., initial location means changing the first sentence. It is also important to mention that since the SST-5 dataset consists of single-sentence reviews, all three locations change the same sentence and thus has the same performance in Figure 2.

As the figure shows, the Sentence-level trigger is able to achieve almost a perfect ASR for all datasets, i.e., it achieves above 97% for Amazon dataset and nearly 100% for the remaining datasets, with a negligible utility loss.

### 3.5. Semantics Consistency Evaluation

BadNL should preserve the semantics of the target input for avoiding the detection. We use a pre-trained **SBERT** from the open-source framework SentenceTransformer (Reimers & Gurevych, 2019) to measure the semantic similarity of our clean, backdoored input pairs. The results show that our techniques achieve a $\widehat{sim}$ score above 0.95, 0.92 and 0.94 for three classes of triggers, which confirms the semantic-preserving property of our techniques.

### 3.6. Comparison of All Attacks

Finally, we compare three different classes of triggers. First, comparing their performance, the Word-level triggers have the best performance, followed by the Sentence-level triggers than the Char-level triggers.

Then, we compare the pros and cons of each class of triggers. The Word-level trigger is the simplest to implement with a fixed trigger, however, the fixed trigger makes it the easiest to detect. The Char-level trigger is more invisible with dynamic words used as triggers for different inputs, however, it may cause a semantic abnormality. Finally, the Sentence-level trigger only converts the tense of the input, which maintains the natural semantics and evades grammar check.

In short, in terms of performance, Word-level trigger comes first followed by Sentence-level one then Char-level trigger. But in terms of invisibility, Word-level trigger comes the last after both of Sentence-level and Char-level triggers.

## 4. Conclusion

In this work, we explore the backdoor attacks against NLP models. We propose three techniques to construct backdoor triggers, namely Word-level, Char-level and Sentence-level. We evaluate our techniques with three datasets. The results show that all three techniques can achieve good attack success rate, while maintaining the utility of the target model.

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
