# OpenReview forum: "BadNL: Backdoor Attacks Against NLP Models"
_ICML.cc/2021/Workshop/AML — ICML 2021 Workshop AML Poster_

### Official Review · Reviewer_TbEK · 2021-06-19
**Have some interesting findings but some important references are missing.**

**Rating:** Accept
**Confidence:** 5

**Review:**

This paper discussed how to backdoor an NLP application, i.e., the sentiment analysis. Specifically, they proposed three different levels of triggers, including the Word-level, Char-level, and Sentence-level triggers. The author also verified their methods on three benchmark datasets.

The paper is well-written and the topic is of great significance and is suitable for this workshop. There are also some interesting discussions, especially using different tenses for the trigger, which is the main reason why I vote for the acceptance. However, I still have some concerns, as follows:


Major Comments
1.	Although there are lots of differences between images and sentences. The proposed approach is still very similar to the BadNets.
2.	Missing important references.
    1) There have been some backdoor attacks (e.g., [1, 2, 3]) against NLP (You can find more on: https://github.com/THUYimingLi/backdoor-learning-resources). The author needs to mention them in related work and even compare some of them in the experiments.

    2) Missing lots of references in the discussion of backdoor attacks. I would suggest the author at least cite some important surveys (e.g., [4, 5]).


Minor Comments
1.	It would be better to cite the journal version of (Gu et al., 2017).
2.	Typo in Line 86-87: `Attack success rate(ASR)’ should be `Attack success rate (ASR)’.


References

[1] A Backdoor Attack Against LSTM-based Text Classification Systems. IEEE ACCESS, 2019.

[2] Poison Attacks against Text Datasets with Conditional Adversarially Regularized Autoencoder. EMNLP-Findings, 2020.

[3] Trojaning Language Models for Fun and Profit. arXiv, 2020.

[4] Data Security for Machine Learning: Data Poisoning, Backdoor Attacks, and Defenses. arXiv, 2020.

[5] Backdoor Learning: A Survey. arXiv, 2020.

---

### Decision · Program_Chairs · 2021-06-21

**Decision:**

Accept (Poster)

**Comment:**

This paper studied backdoor attacks on NLP. The authors can further address the reviewer's comments.